# A Novel Genetic Variant in the WFS1 Gene in a Patient with Partial Uniparental Mero-Isodisomy of Chromosome 4

**DOI:** 10.3390/ijms22158082

**Published:** 2021-07-28

**Authors:** Maurizio Delvecchio, Federica Ortolani, Orazio Palumbo, Concetta Aloi, Alessandro Salina, Francesco Claudio Susca, Pietro Palumbo, Massimo Carella, Nicoletta Resta, Elvira Piccinno

**Affiliations:** 1Metabolic Disease and Genetics Disorders Unit, Giovanni XXIII Children’s Hospital, 70126 Bari, Italy; federicaortolani@hotmail.com (F.O.); lorenzoeluigi@libero.it (E.P.); 2Medical Genetics Unit, IRCCS Casa Sollievo della Sofferenza, 71013 San Giovanni Rotondo (FG), Italy; o.palumbo@operapadrepio.it (O.P.); p.palumbo@operapadrepio.it (P.P.); m.carella@operapadrepio.it (M.C.); 3Clinica Pediatrica, LABSIEM, IRCCS Istituto Giannina Gaslini, 16147 Genoa, Italy; concettaaloi@gaslini.org (C.A.); alessandrosalina@gaslini.org (A.S.); 4Department of Biomedical Sciences and Human Oncology (DIMO), Division of Medical Genetics, University of Bari “Aldo Moro”, 70125 Bari, Italy; francescoclaudio.susca@uniba.it (F.C.S.); nicoletta.resta@uniba.it (N.R.)

**Keywords:** Wolfram syndrome, uniparental disomy, chrosome 4, *wolframin*, syndromic diabetes

## Abstract

Wolfram syndrome is a rare autosomal recessive disorder characterized by optic atrophy and diabetes mellitus. Wolfram syndrome type 1 (WFS1) is caused by bi-allelic pathogenic variations in the *wolframin* gene. We described the first case of WFS1 due to a maternal inherited mutation with uniparental mero-isodisomy of chromosome 4. Diabetes mellitus was diagnosed at 11 years of age, with negative anti-beta cells antibodies. Blood glucose control was optimal with low insulin requirement. No pathogenic variations in the most frequent gene causative of maturity-onset diabetes of the young subtypes were detected. At 17.8 years old, a rapid reduction in visual acuity occurred. Genetic testing revealed the novel homozygous variant c.1369A>G; p.Arg457Gly in the exon 8 of *wolframin* gene. It was detected in a heterozygous state only in the mother while the father showed a wild type sequence. In silico disease causing predictions performed by Polyphen2 classified it as “likely damaging”, while Mutation Tester and Sift suggested it was “polymorphism” and “tolerated”, respectively. High resolution SNP-array analysis was suggestive of segmental uniparental disomy on chromosome 4. In conclusion, to the best of our knowledge, we describe the first patient with partial uniparental mero-isodisomy of chromosome 4 carrying a novel mutation in the *wolframin* gene. The clinical phenotype observed in the patient and the analysis performed suggest that the genetic variant detected is pathogenetic.

## 1. Introduction

Wolfram syndrome type 1 (WFS1; OMIM #222300) is a rare autosomal recessive disorder whose minimal diagnostic criteria are optic atrophy (OA) and juvenile onset non-autoimmune diabetes mellitus (DM). These features may be present in childhood but also later in life, and typically, but not invariably, diabetes mellitus is detected first. It is also known with the acronym of DIDMOAD syndrome as the other pivotal clinical features are usually diabetes insipidus (DI) and progressive sensorineural deafness (D). In addition, urinary tract abnormalities, ataxia, dementia or intellectual disability, and psychiatric illnesses could be observed [1]. From a molecular point of view, Wolfram syndrome is caused by loss of function homozygous or compound heterozygous pathogenic variations in the gene *WFS1*. The gene is mapped to chromosome region 4p16 and encodes for wolframin, a transmembrane protein, which plays a pivotal role in membrane trafficking, secretion, processing, and/or regulation of endoplasmic reticulum (ER) calcium homeostasis [2].

To date and to the best of our knowledge, over 170 different pathogenic variations have been reported and linked to the disease [3]. Missense changes that lead to amino acid substitution represent the most frequent molecular consequence, while nonsense and frameshift are less frequent. Most of the variants are located on exon 8. The clinical course is highly variable, even within the same family, and a genotype–phenotype correlation is a difficult task due to the molecular complexity of Wolfram syndrome and the different clinical characteristics [4,5].

In this manuscript, we report on the first case of WFS1 due to a maternally inherited mutation with partial uniparental mero-isodisomy of chromosome 4.

## 2. Results

Genetic testing revealed the novel homozygous pathogenic variant c.1369A>G; p.Arg457Gly in exon 8 of WFS1 gene (NM_006005.3) (Figure 1). It was found in a heterozygous state in the mother, but it was not detected in the father and has not yet been reported in the literature or in the Exac database. The sibling showed a wild type sequence. In silico analysis by Polyphen 2 indicated this variant as “likely damaging”, while Mutation Tester and Sift classified it as “polymorphism” and “tolerated”, respectively. Following this pipeline, no clinically significant copy number changes were identified, while two wide regions of homozygosity that were suggestive of partial uniparental mero-isodisomy disomy (UPD) have been identified on chromosome 4 (UPD4) (Figure 2).

## 3. Discussion

The clinical diagnosis of Wolfram syndrome is based on the association of DM and OA, while the genetic diagnosis relays on the detection of pathogenic variants in both of the WFS1 gene alleles. We have previously described a very rare case of Wolfram syndrome type 2 [6] and here we report the first case of WFS1 caused by the c.1369A>G; p.Arg457Gly homozygous variant of the *WFS1* gene as a consequence of partial mero-isodisomy of maternal chromosome 4. This novel missense change was found in a homozygous state in the proband and it was absent from databases such as gnomAD, 1000 Genomes and dbSNP. In 2005, Giuliano et al., described in the same position, 457, the substitution of Arginine 457 with another amminoacid the Serine detected in heterozigousity in a WFS1 patient with an atypical phenotype [7]. In our case, the segregation of the genetic variant with the typical phenotype leads us to consider this genetic variant as “likely pathogenic”. The unusual maternal inheritance is related to the partial maternal UPD of chromosome 4. This event is rare but not impossible, causing homozigosity for a genetic variant carried by just one of the parents, and thus confirming the diagnosis of Wolfram syndrome.

Mero-isodisomy may coexist on the same pair of uniparental chromosomes due to one or more recombination events in the prophase of the first meiotic division; afterward, meiotic chromosome segregation errors (non-disjunction at meiosis I or meiosis II, but also reverse segregation and premature separation of chromatids) originate an disomic/nullisomic gamete. The mitotic correction of the resulting aneuploid embryo (trisomy/monosomy rescue) may cause UPD. The UPD phenotypic effects may be due to the involvement of imprinting regions or isodisomy, and can result in a recessive disorder caused by a homozygous pathogenic variant that is present in only one of the parents [8,9].

The SNP array analysis of our patient suggests an MII segregation error of chromosome 4 in maternal meiosis, prior to the meiotic segregation three cross-over events having taken place between the long arms of the homologous chromosomes, then recombinant and non-recombinant sister chromatids are segregated into a disomic egg. A trisomic conceptus is produced following fertilization. The rescue of the trisomy by another error in postzygotic mitosis (paternal chromosome 4 was discarded) results in maternal UPD4 with isodisomy for non-recombinat regions (4p16.3 proximal and 4q28.2q32.3 interstitial) and heterodisomy for recombinant regions (4q12q28.1 interstitial and 4q32.3qter distal). *WFS1* gene variant maps to 4p16.

A few dozen cases of isodisomy unmasking a mutant recessive allele have been reported, even few cases of UPD4 [10,11,12,13,14]. In most cases, the sequence of the events was similar to that described in our case, but also one case of paternal UPD4 [15] and two cases of incidental findings [16,17] have been described.

The incidence of UPD is mainly ascertained through an abnormal phenotype due to an imprinting disorder or recessive disorder, but a recent study on the frequency of UPD in a large population of mostly healthy individuals estimates that the frequency of whole chromosome UPD is approximately 1 in 2000 live births [18]. Maternal UPD is more frequent than paternal UPD and increases with maternal age; however low the risk of UPD, it is necessary to take this into account for genetic counselling as a cause of autosomal recessive disorders, especially when a rare homozygous pathogenic variant is detected.

A few years ago, Papadimitriou et al. [19] described an unusual WFS1 presentation in a patient with maternal UPD4 with a homozygous mutation in the *wolframin* gene. The proband was a 10-year-old girl with type 1 DM diagnosed at 6 years of age, high-frequency sensorineural hearing loss, and a reduction in visual acuity. Her parents and her brother were all healthy. The father and the brother did not carry the defect, which was found in her mother in a heterozygous state. A SNP array analysis for genotyping and a microsatellite analysis to determine the origin of the second allele were performed, showing UPD due to uniparental segregation of the maternal chromosomes. However, our patient carries a mero-isodisomy of maternal chromosome 4, while the patient described by Papadimitrou et al., presented with UPD.

Finally, we would like to report on the parents’ response. We discussed the diagnosis of WFS1 after the genetic testing with them. We asked about their feeling in the case of diagnosis when the patient was 12 years old. They replied that they were happy that the diagnosis had not been conducted before, when she presented only diabetes, because it would have been terrible as blindness could not be prevented. Even if this is a single experience, we are prompted to say that the diagnosis of WFS1 at the pre-clinical/paucisymptomatic stage may warrant so much caution. Early diagnosis is important to enable proper counselling and to prevent complications, but in the absence of any possible treatment to prevent disease progression, the backlash of diagnosing such a neurodegenerative disorder should be evaluated.

## 4. Materials and Methods

Clinical report. The proband, a female, was the first of two siblings born from healthy nonconsaguineous parents. The brother was healthy. None of the relatives developed diabetes mellitus or any kind of eye disease. She was a full term infant born from caesarion section after abruptio placentae. The neurodevelopmental milestones were regular. Menarche occurred at 10 years old. DM was diagnosed at 11 years of age (HbA1c 8.5%—DCA2000, no ketosis). Specific autoantibodies against beta-cells were absent. One year later, the insulin requirement was low (0.29 IU/kg/day) with optimal blood glucose control (HbA1c constantly < 6.5%). Specific autoantibodies for DM were newly assyaed and tested negative. Although the family history was not suggestive of DM or dysglycemia, a “de novo mutation” causative of Maturity-Onset Diabetes of the Young (MODY) was suspected in consideration of the evidence that de novo mutations of *GCK*, *HNF-1 α* and *HNF-4α* may be more frequent in MODY than previously assumed [20]. *HNF-4α*, *GCK*, *HNF-1α*, *PDX1*, *HNF-1β*, and *NEUROD1* were sequenced but no pathogenic variants were detected. Consequently, we suggested that further genetic testing could be useful to investigate the etiology of her DM, but at this stage the patient and her parents decided to not undergo further genetic testing to avoid stress and anxiety. During the follow-up, HbA1c values were constantly below 6.5%, C-peptide levels were in the middle-low normal range and renal function tests were normal. Daily insulin requirement was always low (about 0.2 IU/kg/day), even during puberty. No severe hypoglycemias or ketoacidosis episodes ever occurred.

At 17.8 years old, she presented a rapid reduction in visual acuity and she was admitted to the Ophtalmologist Unit, where optic subatrophy was diagnosed. On ophthalmic examination, her visual acuity was 2–3/10 (right eye) and 1/10 (left eye). Fundus examination showed pale discs in both eyes. Fluorescein angiography and brain MRI were normal except for partial atrophy of eye nerves. She never complained about hearing impariment and audiogram confirmed a normal hearing function. Due to the copresence of DM and optic subatrophy, Wolfram syndrome was suspected and *WFS1* sequencing was requested.

At last clinical evaluation, at 21 years old, the patient was on basal-bolus insulin regimen therapy. Blood glucose control has always been at target (HbA1c < 7%) with a very low insulin requirement (0.18–0.25 IU/kg/day), weight and height 25–50° percentiles; last BMI: 21.7 kg/m^2^. A recent comprehensive ophthalmic examination described worsening of visual acuity (1/10 in both eyes). She is supported by a psychologist.

DNA extraction. DNA from the proband, parents, and her brother was extracted from whole blood using High Pure PCR Template Preparation Kit (Roche, Mannheim, Germany). This family provided written informed consent to molecular testing and to the full content of this publication.

Sanger Sequencing. Exons and flanking regions of *WFS1* were amplified by PCR using previously described primers [21]. Amplicons were purified with exonuclease I and shrimp alkaline phosphatase (ExoSap-IT, USB Corporation, Staufen, Germany) and then sequenced for both sense and antisense strands using an automated fluorescent sequencing method (Big Dye Terminator Kit v1.1, Applied Biosystems, Waltham, MA, USA). The products were separated on an ABI PRISM sequencing apparatus 3730 (Applied Biosystems). All variations were validated by sequencing both DNA strands of three independent PCR products.

SNP-array analysis. High resolution SNP-array analysis was performed on genomic DNA extracted from peripheral blood lymphocytes of the patient, her brother and her parents using the CytoScan HD Array (Thermo Fisher Scientific, Waltham, MA, USA) as previously described [22]. Data analysis was performed using the Chromosome Analysis Suite software version 4.2 (Thermo Fisher Scientific, Waltham, MA, USA). The significance of each of the detected copy number variations (CNVs) was determined by comparing all chromosomal alterations identified in the patient with those collected in an internal database of ~4500 patients studied by SNP arrays since 2010, and public databases including the Database of Genomic Variants (DGV), DECIPHER, and ClinVar. Base pair positions, information about genomic regions and genes affected by CNVs, and known associated disease have been derived from the University of California Santa Cruz (UCSC) Genome Browser, build GRCh37 (hg19). The clinical significance of each of the rearrangements detected has been assessed following the American College of Medical Genetics (ACMG) guidelines [23].

## 5. Conclusions

In conclusion, we describe the first case of Wolfram syndrome due to uniparental maternal mero-isodisomy UPD4 carrying a novel genetic variant, and presenting with diabetes mellitus at 10 years and optic atrophy at 18 years old. Furthermore, this case calls also for caution in testing patients for *wolframin* gene defects when they present only diabetes mellitus, as there is currently no effective therapy to prevent disease progression.

## Figures and Tables

**Figure 1 ijms-22-08082-f001:**
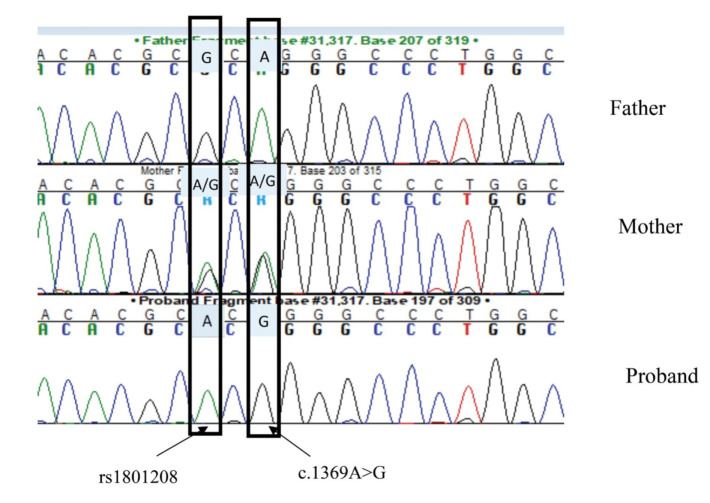
Segregation of WFS1 novel missense variant c.1369A>G; p.Arg457Gly and rs1801208 in family pedigree. Electopherogram shows that proband was the missense variant and SNP were detected in a homozygous state in the proband and in a heterozygous state in the mother. The father was wild type for both alleles.

**Figure 2 ijms-22-08082-f002:**
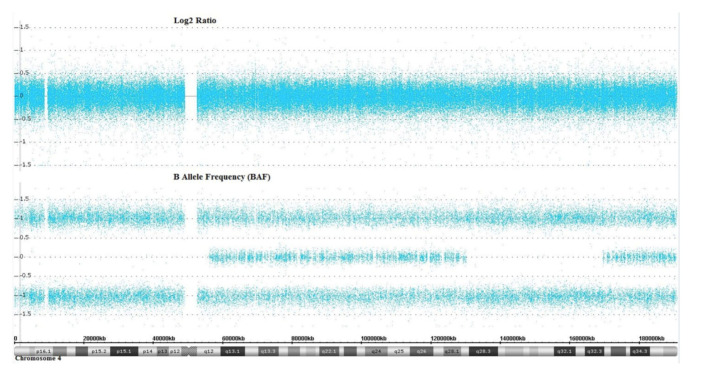
SNP-arrays identified the presence of a segmental UPD on chromosome 4. The log2 ratio indicates a normalized intensity value of each SNP across chromosome 4 compared to diploid individuals. In detail, the log2 ratio of 0 indicates the absence of copy number alterations (i.e., microdeletions and/or microduplications) on chromosome 4 in our patient compared to controls. The B allele can have values of 1 (BB), 0 (AB) and −1 (AA) in a diploid individual. Data show that the patient had a segmental UPD spanning the chromosome 4p and 4q regions. In detail, the B allele frequency shows a 56.1-Mb loss of heterozygosity at chromosome 4p16.3q12 (46,690-56,195,626; hg19), and a 39.4-Mb loss of heterozygosity at chromosome 4q28.2q32.3 (130,191,096-169,630,204; hg19). Genotype analysis confirmed an allelic imbalance compatible with maternal chromosome 4p16.3q12 and 4q28.2q32.3 UPD (data not shown).

## Data Availability

Due to the privacy policy, data are available on request after approval of the patients.

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
