# Peer review of "A Novel Genetic Variant in the WFS1 Gene in a Patient with Partial Uniparental Mero-Isodisomy of Chromosome 4"

_ijms, 2021, doi:10.3390/ijms22158082_

Round 1

Reviewer 1 Report

The authors describe a partial uniparental disomy at chromosome 4 that may be pathogenic and cause of Wolfram syndrome. The manuscript is well written and add to our knowledge about pathogenic genetic mutations that may be related to Wolfram syndrome.

I have the following comments:

  1. Did the authors do DNA sequence analysis of the patient's healthy sibling?
  2. The authors cite a study from Giuliano et al. (2005) that described the same substitution in heterozygous state in a WFS1 patient. The patient's mother of the current study is also heterozygous for the same substitution but she is healthy. Could authors discuss this discrepancy with the Giuliano's study?

Author Response

Thank you for your comments. Please find the point-by-point reply. We checked and corrected some language errors. 

1. the DNA of the proband's sibling resulted normal. This sentence was added to the manuscript.

2. the missense change described by Giuliano et al (Arg457Ser) is different from the variant reported in our manuscript (Arg457Gly). The novel change described in our report, was not previously described or reported in any the genetic database. Basing on the Giuliano’s report and computational analysis, we classify Arg457Gly as likely pathogenic. The our proband’s mother was heterozygous for Arg457Gly and she was healty.

Reviewer 2 Report

This is a straight-forward case report of an individual with typical Wolfram syndrome. They identified a novel mutation in which it looked to be homozygous in the proband but the mutation was only observed in the mother but not in the father. Upon, closer investigation the authors found the proband had inherited uniparental disomy of chromosome 4 from the mother with regions of isodisomy and heterdisomy on the chromosome using snp arrays resulting in an autosomal recessive condition. It is an interesting case report.

Some minor changes include:

There isn’t common usage for the term mero-isodisomy so the term uniparental maternal isodisomy should be substituted.

When the mutation is first described in the text, please add the accession number of the cDNA the numbering of the mutation refers to.

In the figure 1 Instead of putting the genotype next to the sequence the sequence should be labeled as mother, father and proband and discuss the genotypes in the figure legend. There are labels in the sequence panels but the font is so small, it is unlikely it could read clearly. Also, in the figure legend please explain that rs1801208 is a known polymorphism in the gene and is indicated in the figure and it is two basepairs 5’ of the variant that was identified. The polymorphism also shows the proband inherited the mother’s allele and is missing the father’s allele an additional piece of evidence of isodisomy.

Figure 2 legend – not sure what this means “(ratio 1 is log2 ratio of 0)” and how it pertains to the figure.

The last paragraph of the discussion and the last sentence of the conclusions should be removed. The authors come to this conclusion based on the feelings of one set of parents which are valid for them but a wider study as to how parents of wolfram syndrome children and their diagnoses should be performed if the authors want to generalize as to when molecular diagnosis for this disorder or other disorders that do not have a treatment should be performed. Other parents may have different opinions.

There are a number of grammatical error and unfortunately, I started down the rabbit hole of noting them so there are several changes to be made here.

Grammatical changes

Please italicize gene names

Please check for spelling errors

Line 18, please add a “the” before “wolframin gene”

Line 42, please change mapped “on” to mapped “to”

Line 46, please change “Up to date,” to “To date and”

Line 48 amino acid is two words not one

Line 53 please change the word “maternal” to “maternally”

Line 57 please “the” before exon 8

Line 58 and 59 Please change “has not yet reported in literature neither in Exac” to “has not yet been reported in the literature or in the Exac”

Line 83 please add the word “the” in front of “WFS1 gene”

Line 85, please delete “at” and replace it with “in a” and delete “the” before database and change “database” to “databases”

Line 87 insert an “an” before atypical

Line 94, the “o” should be an “or”

Line 103, insert “events” after cross-over

Line 104, add a “the” in front of long arms and insert “the homologous” in front of “chromosome” and make chromosome plural.

Line 105, delete “in” and replace with “into a”

Line 110, replace “on” with “to”

Line 115, replace “to” with “an”

Line 118, replace “of” with “is” and replace “birth” with live births.

Line 122, add an “A” to the beginning of the sentence

Line 123, replace “She” with “The proband”

Line 124, replace “since” with “diagnosed at”

Line 127, replace “at” with in and “an” with “a”

Line 129, replace “alleles” with “chromosomes”

Line 131, replace “an” with “with”

Line 146, nonconsaguineous is one word.

Author Response

Thank you for your comment. Please, here you can find our poin-by-point reply.

  1. Sorry about that. A common usage for the term mero-isodisomy was adopted in the manuscript.
  2. we add the accession number of the cDNA of WFS1 as requested.

  3. Figure 1 and legend were modified as requested

  4. The legend of figure 2 was modified as requested
  5. We modified the last paragraph. De decided to not eliminate the whole paragraph, we thought that shortening the paragraph and "softening" the message could be helpful for readers. We hope that the reviewer agree with this point. If it does not agree with us, the paragraph will be eliminated. 
  6. Thank for the suggestions. We corrected all the errors.